# Ultrastrong coupling between nanoparticle plasmons and cavity photons at ambient conditions

Denis G. Baranov [1], Battulga Munkhbat [1], Elena Zhukova[2], Ankit Bisht[1], Adriana Canales [1],
Benjamin Rousseaux[3], Göran Johansson[3], Tomasz J. Antosiewicz [1,4] & Timur Shegai [1 ✉]

Ultrastrong coupling is a distinct regime of electromagnetic interaction that enables a rich variety of intriguing physical phenomena. Traditionally, this regime has been reached by coupling intersubband transitions of multiple quantum wells, superconducting artificial atoms, or two-dimensional electron gases to microcavity resonators. However, employing these platforms requires demanding experimental conditions such as cryogenic temperatures, strong magnetic fields, and high vacuum. Here, we use a plasmonic nanorod array positioned at the antinode of a resonant optical Fabry-Pérot microcavity to reach the ultrastrong coupling (USC) regime at ambient conditions and without the use of magnetic fields. From optical measurements we extract the value of the interaction strength over the transition energy as high as $g/\omega \sim 0.55$, deep in the USC regime, while the nanorod array occupies only ~4% of the cavity volume. Moreover, by comparing the resonant energies of the coupled and uncoupled systems, we indirectly observe up to ~10% modification of the ground-state energy, which is a hallmark of USC. Our results suggest that plasmon-microcavity polaritons are a promising platform for room-temperature USC realizations in the optical and infrared ranges, and may lead to the long-sought direct visualization of the vacuum energy modification.

---

[1] Department of Physics, Chalmers University of Technology, 412 96 Göteborg, Sweden. [2] Moscow Institute of Physics and Technology, Dolgoprudny, 141700 Moscow, Russia. [3] Department of Microtechnology and Nanoscience—MC2, Chalmers University of Technology, 412 96 Göteborg, Sweden. [4] Faculty of Physics, University of Warsaw, Pasteura 5, 02-093 Warsaw, Poland. ✉email: timurs@chalmers.se

Two coupled harmonic oscillators is one of the most basic physical toy models that can be employed to understand the behavior of various mechanical and electromagnetic systems in simple intuitive terms. Usually, this approach is described by Newton's equations of motion for the oscillators' amplitudes $x_1$ and $x_2$, wherein the coupling is mediated by the bi-linear interaction term ($\propto g x_1 x_2$ or $g(x_1 - x_2)^2$ depending on the system, where $g$ is a coupling constant) in the system's Hamiltonian[1]. Such a mechanistic approach has been widely used to model the optical response of coupled plasmonic nanoparticles[2], exciton–polaritons[3,4], plasmon–excitons[5,6], and magnon–polaritons[7,8]. While in the weak or strong coupling regime, $g \ll \omega$, the validity of this approach is accepted, it might not provide an adequate description of coupled electromagnetic systems when the coupling constant reaches a considerable fraction of the resonance energy, that is, $g \sim \omega$.

Ultrastrong coupling (USC) is a regime of light–matter interaction in which the coupling strength, $g$, exceeds about 10% of the transition energy, $\omega$[9,10]. In this regime, the standard quantum optical approximations, such as the commonly made rotating wave approximation (RWA), fail. Thus so-called fast-rotating terms, as well as the quadratic $A^2$ term must be taken into account in order to correctly describe the system's behavior[11–13]. The latter arises from the expansion of the minimal coupling Hamiltonian $\left(\mathbf{p} - \frac{e}{c}\mathbf{A}\right)^2$ (where $\mathbf{p}$ and $\mathbf{A}$ are the particle's momentum and the field's vector potential, respectively) and is absent in the naive coupled oscillators model. Remarkably, not only quantum two-level systems, but also classical harmonic oscillators in the regime of ultrastrong coupling require description using the full Hamiltonians[14]. One of the intriguing implications of USC is that the global vacuum energy of the system becomes dependent on the coupling strength[15]; that is, if the coupling constant is allowed to vary, it will cost a certain amount of energy to adjust the value of the coupling strength. Additionally, the ground state gains a photonic component, that is, it contains a finite amount of virtual photon excitations[15,16]. This in turn may lead to highly unusual phenomena, such as dynamical Casimir effect[17–19] and single-photon frequency conversion[20].

Although the USC domain of light–matter interaction is of significant fundamental interest, it remains largely unexplored experimentally due to technical challenges of its realization. Indeed, so far the record-high realizations (where $g/\omega > 1$) have been based on Landau polaritons[21] and superconducting circuits[22], which require cryogenic temperatures and high magnetic fields. This specific interaction regime for which $g/\omega > 1$ is called "deep" strong coupling. Replicating such results under ambient conditions remains a challenge. Room temperature realizations using collective coupling of organic molecules with microcavities have reached $g/\omega$ of "only" $\sim 0.3$[23,24], with the recent implementation based on intersubband transitions of doped quantum wells showing $g/\omega \sim 0.7$[25]. However, such values are usually achieved by totally saturating the cavity volume with the material. Plasmonic lattices[26,27] as well as single plasmonic nanorods[28] have been shown to couple strongly with microcavity modes previously, however, the reported interaction strengths have not reached the level of the USC regime.

Here, we use our recently developed strategy based on plasmon–microcavity polaritons[29] to achieve considerably higher coupling strengths, well into the USC regime, at room temperature. The plasmon–microcavity polaritons employed in this study consist of densely packed plasmonic nanorod arrays fabricated at the antinode of the Fabry–Pérot microcavity formed by two gold mirrors. By fitting the experimental reflection data by the spectrum of the full Hopfield Hamiltonian, we extract the normalized coupling strength $g/\omega$ as high as 0.55, one of the highest values for

room temperature realizations of USC. We stress that such a high value is achieved here for a cavity, whose occupied volume amounts to only about 4%, with just a single layer of plasmonic nanoparticles. This makes a clear difference with respect to organic dyes[23] and intersubband transitions[25], which reach similar numbers only by filling nearly 100% of the cavity interior. Furthermore, the experimental data allow us to indirectly observe the modification of the vacuum energy induced by USC, as well as estimate the photonic occupancy of the new ground state. In contrast to coupling bulk quantum wells or 2D electron gases, this vacuum energy effect can be potentially observed directly via the action of the vacuum energy landscape on a discrete nanoparticle. We thus argue that the large oscillator strength of plasmonic nanoparticle arrays, as well as control over their geometrical parameters and density, makes the plasmon–microcavity polaritons studied here an attractive platform for further investigations of room temperature ultrastrong and deep strong coupling regimes.

## Results

**Ultrastrong coupling in plasmon–microcavity systems**. The system under study is illustrated in Fig. 1a. It consists of a sub-diffractive periodic array of gold (Au) nanorods placed at the antinode of the fundamental Fabry–Pérot (FP) microcavity mode formed by two Au mirrors and filled by a $SiO_2$ spacer. The nanorod array couples to the vacuum field of the FP microcavity, thus producing plasmon–cavity polaritons manifested as distinct resonant spectral features emerging in transmission, reflection, and absorption spectra of the coupled system.

To provide initial insight into the behavior of the coupled system, we perform numerical finite-difference time-domain (FDTD) simulations (FDTD Solutions, Lumerical). Figure 1b shows a map of absorption spectra of coupled FP–nanorod systems at normal incidence with the electric field parallel to the nanowires as a function of the cavity thickness for nanorod lengths $L = 300$ nm and $dy = 30$ nm spacing in the $y$-direction. For an easy comparison between these coupled system spectra with the uncoupled elements, we plot the bare FP cavity resonances with curved lines. The vertical dashed line marks the bare plasmon nanorod resonance of the array. A comparison clearly shows a rather complicated picture of new eigenmodes' dispersion in which the even FP modes are practically unperturbed while the odd FP modes are shifted significantly from the bare cavity positions.

The 1st order FP mode of an empty cavity intersects the bare nanorod array plasmon resonance around 400 nm cavity thickness resulting in a distinct anticrossing (Fig. 1b). The lower polariton (LP) transitions from a plasmon-dominated mode (for a thin cavity) to an FP-dominated mode at large detuning (for a thick cavity). However, the upper polariton (UP) upon acquiring a plasmon-like character at large detuning, crosses the 2nd order FP mode and approaches the spectral position of the 3rd FP cavity mode, which in the coupled system is strongly pushed to the blue due to hybridization with the plasmon. Such qualitative behavior is observed for all the odd FP modes: each odd coupled $i$th mode is pushed to the blue beyond the subsequent even $i + 1$st mode (which is unperturbed) and approaches the $i + 2$nd odd FP mode. In fact, at no point in the spectral analysis do any of the FP–plasmon–polaritons follow the plasmon dispersion. In contrast, the even modes in the coupled system do not significantly interact with the array because they have a node of the electric field in the center of the cavity, where the rods are positioned. These observations suggest that a multimode character of the FP microcavity is important for a detailed interpretation of our results.

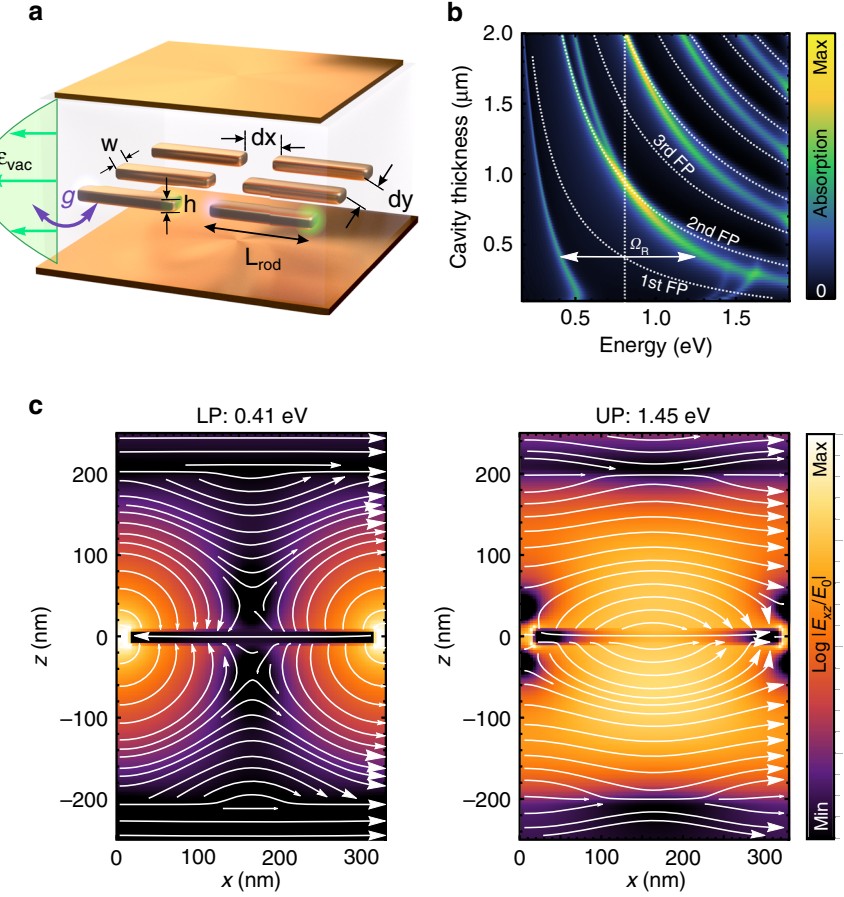

**Fig. 1 Sketch and numerical modeling of the coupled system. a** Artistic illustration of the system: an array of plasmonic nanorods positioned in the middle of a Fabry–Pérot cavity formed by two gold mirrors. The cavity interior is filled with SiO₂. The array couples to the FP cavity mode, exchanging energy at a rate *g*. **b** False-color normal-incidence absorption spectra as a function of cavity thickness with an array of 300 nm long plasmonic nanorod (width 50 nm, height 20 nm) positioned in the middle of SiO₂-filled Fabry–Pérot cavity. The vertical dashed line indicates the nanorod plasmon resonance outside of the cavity. The curved lines indicate resonances of the empty FP cavity, whose even modes are not modified by the coupling. $\Omega_R$ denotes plasmon–cavity mode splitting at zero detuning. **c** The electric field intensity (in the log scale) and the electric field lines in the vertical plane across the middle of the nanorod induced by a normally incident plane wave (polarized in the figure plane) for the coupled system of 400 nm thick cavity and 300 nm long nanorods calculated for the lower and upper polaritons.

Another remarkable feature of the absorption map in Fig. 1b is the dispersion of the lower polariton in the thin cavity limit: for cavities thinner than about 200 nm, the LP dispersion exhibits a back-bending to extremely low energies, Fig. 1b, which is likely related to the near-field interaction of discrete plasmonic nanoparticles with the cavity mirrors in this short-range limit. This behavior is not reproduced by the Hamiltonian modeling, and we will not consider it in detail in the following.

The spatial distributions of the electric field induced by a normally incident plane wave inside the plasmon–cavity system calculated at the resonant energies for a 400 nm thick cavity (Fig. 1c) clearly display the opposite symmetries of the two resonances. While the lower energy mode shows an anti-symmetric combination of cavity and plasmon fields, featuring two saddle points above and below the nanorod, the upper energy mode is a symmetric combination. Such behavior highlights the polaritonic nature of the two resonances of the hybrid system. For a 400 nm thick cavity, corresponding to near-resonant coupling ($\omega_{cav} = \omega_{pl} \sim 0.8$ eV), the Rabi splitting, $\Omega_R$, estimated as the energy difference between the two absorption peaks reaches ~1 eV. Thus, assuming that $\Omega_R = 2g$ on resonance, we estimate the normalized coupling strength of $g/\omega_{pl} > 0.5$, which clearly indicates the ultrastrong coupling regime in the system. In what follows, we perform a more

rigorous estimation of the $g/\omega_{pl}$ values in our systems based on a full Hopfield Hamiltonian.

The same qualitative behavior is observed for diluted arrays and ones with longer nanorods, as shown in Supplementary Fig. 1. Although the coupling strength decreases with a smaller nanorod density, the spectral behavior indicates that all the odd modes mix with the plasmon, yielding a complex polaritonic system. However, for longer rods and/or diluted arrays, in addition to the plasmon, higher-order resonances are observed, which are associated with lattice modes. Additional data, including transmission and reflection spectra and complete data for the incident light polarization perpendicular to the nanorod axis, as well as the uncoupled cavity and array elements, are provided in Supplementary Figs. 1–5.

Samples of coupled plasmon–microcavity systems were fabricated by combination of electron beam evaporation (Au mirrors), plasma-enhanced chemical vapor deposition (dielectric spacers), and electron beam lithography (nanorod arrays) (see "Methods" for details). Figure 2a shows a bright-field optical microscope image of the fabricated nanorod arrays with lengths ranging from 200 to 400 nm with a step of 50 nm. The nanorods have fixed height of $h = 20$ nm and width of $w = 50$ nm. An exemplary scanning electron microscope (SEM) image of gold nanorods array with length of $L_{rod} = 250$ nm is shown in Fig. 2b

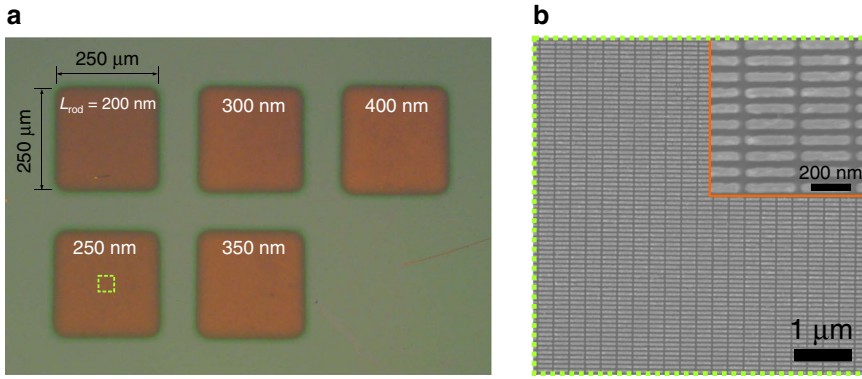

**Fig. 2 Fabricated samples. a** Bright-field optical microscope images of gold nanorod arrays positioned in the middle of a SiO$_2$-filled FP cavity (without the top mirror) fabricated by electron beam lithography. Individual nanorods have a fixed height of $h = 20$ nm, width of $w = 50$ nm, and length varying from 200 to 400 nm. The side-to-side distance between the nanorods is 30 nm. The arrays are 250 × 250 μm$^2$. **b** SEM image of the $L_{rod} = 250$ nm nanorods array. The inset shows a magnified view of the nanorod array.

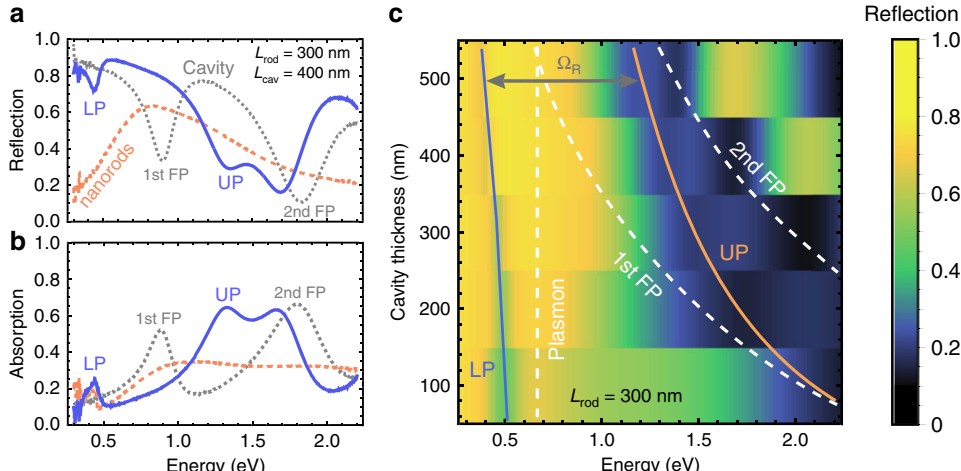

**Fig. 3 Measurements of the fabricated samples. a, b** Measured reflection (**a**) and absorption (**b**) spectra of an empty $L_{cav} = 400$ nm cavity, bare $L_{rod} = 300$ nm long plasmonic nanorods, and those of the coupled system with the electric field polarization parallel to the major rod axis. **c** Measured dispersion of the reflection spectra of the coupled plasmon–cavity system with $L_{rod} = 300$ nm plasmonic nanorods as a function of the cavity thickness revealing an anti-crossing between the two polaritonic modes. Dashed lines show the positions of the bare plasmon array mode and the bare Fabry–Pérot modes of the 1st and 2nd order. $\Omega_R$ denotes the minimal observed splitting between the two polaritonic modes.

(see "Methods" for details). Both the figures clearly show high-density plasmonic arrays with an interparticle distance as small as 30 nm, corresponding to the surface filling factor of 60%. More examples are shown in Supplementary Fig. 6.

Next, we proceed to optical measurements of the fabricated plasmon–cavity systems using Fourier transform infrared (FTIR) spectroscopy (see "Methods" for the details of measurements). Figure 3a, b shows exemplary, measured at normal incidence, reflection, and absorption spectra of an empty 400 nm thick cavity, 300 nm long nanorods array, and those of the coupled system (see Supplementary Figs. 7 and 8 for measured reflection and absorption spectra of all uncoupled cavities and nanorods). The uncoupled cavity and array resonances overlap spectrally and, when coupled, unambiguously confirm the realization of a giant Rabi splitting in the spectra of the coupled plasmon–cavity systems. We note that the 2nd order Fabry–Pérot mode redshifts in the hybrid system although it cannot interact with the nanorod array located exactly in the middle of the cavity. This behavior is consistently observed for all measured coupled systems, Supplementary Fig. 9. It could be explained by the hybrid cavities having slightly larger thickness due to the presence of the plasmonic array that could redshift all the uncoupled Fabry–Pérot modes.

Dispersion of measured normal-incidence reflection spectra from coupled systems with 300 nm long nanorods and varying cavity thickness displays a clear anticrossing between the 1st order Fabry–Pérot mode and the plasmon mode of the array, Fig. 3c (see Supplementary Fig. 9 for dispersions of reflection and absorption spectra vs cavity thickness for all nanorod lengths). The spectra also reveal the 2nd order Fabry–Pérot mode (third dip from the left), which does not interact with the nanorods due to the electric field node in the center of the cavity. As revealed by the Hamiltonian analysis of the spectra in the next section, the nanorods array mode additionally redshifts from ~0.8 to ~0.7 eV due to the presence of a dielectric medium around the rods. For the thinnest 100 nm thick plasmon-loaded cavities, neither reflection nor absorption spectra show the exact position of the upper polariton, which was beyond the detection range of the FTIR microscope used for this set of measurements. For this reason, we performed additional reflectivity measurements in the visible range for the 100 nm thick samples using a normal optical microscope to capture the spectral feature of the upper polariton (see Supplementary Fig. 10). Additional reflection spectra at normal incidence in the visible range were collected using a 20× objective (Nikon, NA = 0.45), directed to a fiber-coupled

spectrometer and normalized with reflection from a standard dielectric-coated silver mirror. Based on these spectra, the vacuum Rabi splitting taken as the energy difference between the two reflection dips at zero detuning ($\omega_{cav} = \omega_{pl}$, 500 nm thick cavity), reaches ~0.8 eV at the resonant energy of ~0.7 eV, Fig. 3c. Thus, the Rabi splitting in our samples exceeds both the bare cavity and bare plasmon resonance frequencies, indicating that the hybrid plasmon–cavity system is deep into the USC regime.

**Analysis of the ultrastrong coupling using Hopfield Hamiltonian.** We now turn to a more thorough analysis of the experimental data. Since a rough estimation already reveals that the Rabi splitting in our system is comparable to the transition energy of uncoupled oscillators, the usual Jaynes–Cummings or Rabi-type coupled Hamiltonians are invalid, and a more general Hamiltonian must be used. Therefore, to analyze our system we employ the full Hopfield Hamiltonian including both the fast-rotating and the quadratic $A^2$ terms, which capture the essential physical characteristics of an ultrastrongly coupled system[9]. We will focus on the two lowest modes of the plasmon–cavity structure, hence we will consider only coupling of two oscillators: the 1st order normal incidence FP mode of the cavity with energy $\hbar\omega_{cav}$, and the collective long-axis plasmon mode of the array with energy $\hbar\omega_{pl}$. Here, the cavity mode plays the role of the light component of the system, whereas the plasmonic nanorod array mode plays the role of the matter component. The total Hamiltonian thus reads:

$$\hat{H} = \hbar\omega_{cav}\left(\frac{1}{2} + \hat{a}^\dagger\hat{a}\right) + \hbar\omega_{pl}\left(\frac{1}{2} + \hat{b}^\dagger\hat{b}\right) + \hat{H}_{int}, \quad (1)$$

where $\hat{a}$ and $\hat{b}$ are the microcavity and collective plasmon annihilation operators, respectively, and $\hat{H}_{int}$ is the interaction Hamiltonian. If we were to consider individual nanoparticle plasmons interacting with each other instead of the collective array mode, the Hamiltonian would also yield additional eigenstates weakly interacting with light[16]. As long as we work away from the Rayleigh modes of the array[30], which is ensured by sub-diffraction periodicity, all the plasmon–plasmon interaction effects can be absorbed into the collective plasmon frequency $\omega_{pl}$.

The interaction part can be written differently depending on the gauge in which the electromagnetic field is treated. The two options that are often used are the Coulomb gauge and its dipole representation. The latter can be obtained from the Coulomb gauge by performing the Power–Zienau–Woolley transformation[31]. When a cavity couples to a two-level system, the two representations are not gauge-invariant because of the two-level approximation[32,33]. However, since we are considering coupling of two harmonic oscillators, the two pictures provide identical

spectra[14,16]. We will therefore use the Coulomb gauge, in which the single-mode interaction Hamiltonian can be written as[34,35]:

$$\hat{H}_{int} = \hbar g_C\left(\hat{a}^\dagger + \hat{a}\right)\left(\hat{b}^\dagger + \hat{b}\right) + \frac{\hbar g_C^2}{\omega_{pl}}\left(\hat{a}^\dagger + \hat{a}\right)^2, \quad (2)$$

where $\hbar g_C = \mu_{pl}\sqrt{a^2\rho}\,\mathcal{E}_{vac}\sqrt{\frac{\omega_{pl}}{\omega_{cav}}}$ is the coupling strength with $\mu_{pl}$ being the transition dipole moment of the plasmonic nanorod, $\rho$ the plasmonic nanoparticles density per unit area $a^2$ ($\sqrt{a^2\rho}$ thus has a familiar $\sqrt{N}$ scaling), and $\mathcal{E}_{vac} = \sqrt{\frac{\hbar\omega_{cav}}{2\varepsilon\varepsilon_0 a^2 L_{eff}}}$ the vacuum electric field of the cavity with $L_{eff}$ being the effective cavity mode transverse thickness[16]. The first term in Eq. (2) is the usual Rabi-type interaction including both slow and fast-rotating terms. The second term is the so-called $A^2$ term, which arises from the expansion of the minimal coupling Hamiltonian $\left(\mathbf{p} - \frac{e}{c}\mathbf{A}\right)^2$ and "protects" the coupled system from the superradiant phase transition[12,13], as well as stabilizes the spectrum against the square-root singularity[36]. Supplementary Figure 12 shows the spectrum of Hamiltonian (1) for $\omega_{pl} = \omega_{cav} = 1$ eV as a function of the coupling constant $g_C$; it also demonstrates that neglecting the $A^2$ term as well as fast-rotating terms leads to incorrect and unphysical spectra.

We want to emphasize that although our system is essentially classical, we choose to use the quantum Hamiltonian because it provides a convenient description in terms of the modes amplitudes via creation operators from start. Moreover, the use of the quantum Hamiltonian allows us to obtain the characteristics of the ground state of the system in a straightforward way, which we analyze in the following. However, the linear response and the energy spectrum of the system can be equally obtained from a classical description not involving any operator algebra, which is demonstrated above by the FDTD simulations.

In a classical optical experiment, the outcome of which is some response function of the system, such as elastic scattering, reflection, or absorption, one cannot access directly the ground-state energy. However, spectral positions of the resonant features in reflection or absorption spectra reflect approximately the transition energies between the ground and first excited states of the system $\hbar\omega_{\pm} = E_{\pm 1} - E_0$. Therefore, to model the system with the Hopfield Hamiltonian framework, we fit the measured dispersions of reflection dips with calculated transition energies $\hbar\omega_{\pm}$ of the Hopfield Hamiltonian[34].

The resulting Hamiltonian fit of a coupled system's resonant transitions as a function of the bare cavity energy is presented in Fig. 4a for $L_{rod} = 300$ nm nanorod arrays. For each cavity thickness, the bare cavity energy was determined from the spectral position of its reflection dip (Supplementary Fig. 7). By

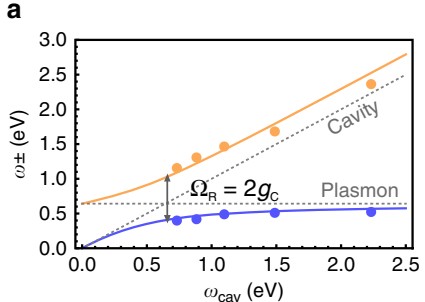
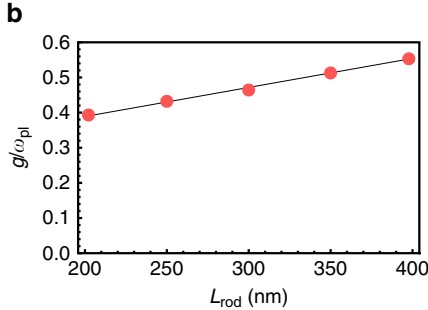

**Fig. 4 Analysis of the experimental data. a** Fitting of the measured polaritonic dispersion of the coupled plasmon–cavity system ($L_{rod} = 300$ nm) with Hopfield Hamiltonian transition energies. Dots show resonant energies of the coupled system extracted as experimental reflection dips, lines are Hopfield polaritons dispersion, gray dashed lines are the bare cavity and bare plasmon energies. $\Omega_R$ denotes Rabi splitting between the two polaritonic modes at the zero-detuning point. **b** Normalized coupling strength $g_C/\omega_{pl}$ at zero detuning versus nanorod length. Solid line is the linear interpolation.

assuming that the effective cavity thickness scales as $L_{\text{eff}} = \frac{\lambda_{cav}}{4n}$ with $n$ being refractive index of the cavity medium, we arrive at the coupling strength in the Coulomb gauge $g_C = \omega_{pl}\mu_{pl}\sqrt{\frac{\hbar\rho}{\pi\varepsilon_0 nc}}$, which is independent of the cavity's thickness and energy. Hence, we fit the polaritonic dispersion by freely varying plasmon frequency $\omega_{pl}$ and the coupling strength $g_C$. For the $L_{\text{rod}} = 300$ nm nanorod arrays, the fitting yields the plasmon frequency of 640 meV and the coupling strength of 300 meV, resulting in Rabi splitting of exactly $2g_C = 600$ meV at resonance, $\omega_{pl} = \omega_{cav}$ (see Supplementary Fig. 13 for Hamiltonian fits of other coupled systems, and Supplementary Table I for extracted plasmon energies and coupling strengths). For all five nanorod lengths, we consistently obtain normalized coupling strength values $g_C/\omega_{pl}$ in the range from 0.4 to 0.56, Fig. 4b, which unambiguously indicate the USC regime of interaction between the nanorods and the cavity modes[9]. Furthermore, we notice that the normalized coupling strength $g_C/\omega_{pl} = \mu_{pl}\sqrt{\frac{\hbar\rho}{\pi\varepsilon_0 nc}}$ is a function of the plasmon transition dipole moment and the particle density only. Therefore, if the product $\mu_{pl}\sqrt{\rho}$ grows with increasing nanorod length, we may expect even higher values of $g_C/\omega_{pl}$ for longer rods resonating at lower energies.

We also compare the resulting fits with those obtained by applying the multimode Hopfield Hamiltonian accounting for all the normal-incidence modes of a Fabry–Pérot cavity, which can be solved analytically[35] (Supplementary Note 1). The most prominent difference is that in the multimode picture the upper polariton crosses the bare plasmon energy exactly at the point where it also crosses the even cavity mode (see Supplementary Fig. 14), in perfect agreement with FDTD simulations (Fig. 1b). Nevertheless, the resulting fits and coupling strengths are very close to the results of the single-mode Hamiltonian analysis (see Supplementary Table II), which justifies its validity in our case. Of course, the single-mode Hamiltonian only yields correct spectra as long as all higher-order (odd) cavity modes are detuned from the array mode, which is the case in our experimental configuration. The multimode Hamiltonian picture, however, may become important in other cases, as was also mentioned in the discussion of numerical FDTD results in Fig. 1b.

It is further instructive to compare the obtained values with an estimation for $g_C$ that can be deduced directly from the geometry of the system. The vacuum electric field can be calculated as $\mathcal{E}_{vac} = \sqrt{\frac{\hbar\omega_{cav}}{2\varepsilon\varepsilon_0 a^2 L_{\text{eff}}}}$ where for the effective cavity mode thickness one can use $L_{\text{eff}} = \frac{\lambda_{cav}}{4n}$. The plasmon transition dipole moment can be estimated from the scattering cross-section of a single nanorod in free space (see Supplementary Note 2) and by applying the classical Larmor formula for the decay rate of a point dipole[37]. This yields the value of around $3.4 \times 10^4$ Debye for the 400 nm long Au nanorod (corresponding to the radiative decay rate of ~67 meV, see Supplementary Table III). Combining these values with the nanorod density $\rho = (430\,\text{nm}\cdot80\,\text{nm})^{-1}$, one obtains the resonant ($\omega_{pl} = \omega_{cav}$) coupling strength of around $g_C \approx 0.3$ eV, which agrees perfectly with the results of fitting.

We further illustrate the importance of keeping the quadratic term by analyzing the data with simpler, albeit a priori incorrect, Hamiltonians. An attempt to fit the experimental with eigenvalues of Hopfield Hamiltonian without the $A^2$ term does not yield any reasonable result with a region of the energy spectrum becoming imaginary, Supplementary Fig. 17, and slightly overestimated coupling strengths. This imaginary spectrum is a fundamental property of the coupled oscillators Hamiltonian without any kind of quadratic stabilizing term[36,38]. Fitting the data with no $A^2$ Hopfield Hamiltonian under RWA (i.e., also without fast-rotating terms), although seems to give a better fit, yields regions with negative LP energy spectrum and largely overestimated coupling strength, Supplementary Fig. 17.

Besides dispersions of polaritonic energies, a remarkable feature of all measured reflection spectra is that the lower (and even upper) polaritons are much narrower than the bare plasmon mode (see, e.g., Fig. 3a and Supplementary Fig. 9). This behavior contrasts the non-Hermitian Jaynes–Cummings and Rabi Hamiltonians, where the decay rates of the photonic mode and electronic transition are "shared" equally at zero detuning between lower and upper polaritons: $\text{Im}\,\omega_\pm = -(\gamma_{cav} + \gamma_x)/4$, where $\gamma_x$ is the linewidth of the electronic transition[3,4]. A similar suppression of the polariton linewidth has been observed with a cyclotron resonance coupled to a Fabry–Pérot cavity[39]. While we cannot directly introduce a non-Hermitian part into Hamiltonian (1), since it will render the ground-state energy complex-valued, we can qualitatively describe the decay processes in the coupled system. As we showed above, the linewidth of a single plasmonic nanorod outside the cavity is largely determined by its radiative loss. For nanorod arrays outside the cavity the radiative decay is even more dominant, reaching ~95% of the total linewidth (see Supplementary Note 2). However, when the dipolar oscillator, such as our plasmonic array, is placed between the mirrors, its radiation does not instantaneously leave the system—instead, it first bounces between the mirrors, and leaves the system only at the cavity's leakage rate $\gamma_{cav}$. Therefore, fast radiative decay of the plasmon array mode becomes irrelevant, and the polariton linewidths are mostly determined by the total cavity's and non-radiative array's decay rates. This qualitative argument explains the apparent narrowness of the observed polaritonic bands. For a more rigorous description of the polaritons linewidth in the ultrastrongly coupled system (which is outside the scope of this work), the master equation approach might be needed[40].

**Ground-state energy and photonic occupancy.** Having performed the fitting of the experimental data, we can analyze how the ground state of the system $|G\rangle$ is modified by the ultrastrong coupling. Again, we will restrict ourselves to the single-mode model of the system. In the uncoupled case, the global ground state is a direct product of the zero-photon and zero-plasmon states $|G\rangle = |0_{cav}\rangle \otimes |0_{pl}\rangle$, and the energy of this state is $E_G = \langle 0|H_{cav} + H_{pl}|0\rangle = \frac{\hbar}{2}(\omega_{cav} + \omega_{pl})$, correspondingly. The USC modifies the global ground state $|\tilde{G}\rangle$ by admixing the states with different number of excitations, i.e., the global ground state with the higher excited states[15], thus modifying the ground-state energy. Since after diagonalization, the coupled system comprises two new harmonic oscillators, its ground-state energy is $\tilde{E}_G = \frac{\hbar}{2}(\omega_+ + \omega_-)$.

The $A^2$ term modifies not only the energies of the polaritonic excited states[41], but also the ground-state energy of the system. Since the ground-state energy of a harmonic oscillator (or a set thereof) is half the transition energy (sum of those), its modification can be calculated as $\delta E_G = \tilde{E}_G - E_G = \frac{\hbar}{2}(\omega_+ + \omega_- - \omega_{cav} - \omega_{pl})$. By expanding the solution of Eq. (6) (see "Methods") into a Taylor series near $g_C = 0$, the ground-state energy modification can be approximated by:

$$\delta E_G = \frac{\omega_{cav}}{\sqrt{2}\omega_{pl}}\frac{\sqrt{A+B}-\sqrt{A-B}}{B}g_C^2 + O(g_C^4), \qquad (3)$$

where we notated $A = \omega_{cav}^2 + \omega_{pl}^2$ and $B = |\omega_{cav}^2 - \omega_{pl}^2|$. At zero detuning ($\omega_{pl} = \omega_{cav}$) this expression yields $\delta E_G = \frac{g_C^2}{2\omega_{cav}} + O(g_C^4)$.

The ground-state energy change at zero cavity–plasmon detuning can be estimated as $\delta E_G \approx \frac{g_C^2}{2\omega_{cav}}$, which for $g_C/\omega_{pl} \approx 0.5$ becomes roughly $\delta E_G \approx g_C/4 \approx 75$ meV accounting for about 12% of the unperturbed ground-state energy $E_G$. This value of the relative energy modification is smaller than what could be obtained in the system studied in ref. [25] with $\frac{g_C}{\omega_0} \approx 0.73$. However, our absolute value is much greater because our system exhibits interacting resonances in the near-IR to visible range with resonant energies around 1 eV, while the characteristic energies of the system in ref. [25] lie 1 order of magnitude lower at around 100 meV (additionally the plasmonic nanoparticle array in our case fills only about 4% of the cavity interior, as opposed to ref. [25] where the active material fully saturates the mode volume). Thus, the absolute ground-state energy change in our system is several times greater than $k_B T$ at room temperature. This, in turn, implies that such ground-state energy modification might be important in practice and may show up in realistic USC-related effects even at room temperature.

The normalized ground-state energy variation $\frac{\delta E_G}{E_G} = \frac{\tilde{E}_G - E_G}{E_G} = \frac{\omega_+ + \omega_-}{\omega_{cav} + \omega_{pl}} - 1$ calculated using the obtained coupling strengths and analytical expressions for polariton energies $\omega_\pm$, Fig. 5a, predicts up to ~10% modification of the ground-state energy for normal incidence Fabry–Pérot mode upon coupling with the plasmonic array (see Supplementary Fig. 18 for the ground-state energy modification for other nanorod lengths). We want to emphasize that this vacuum energy is not an arbitrary reference level for all the higher energy states of the system. If the coupling constant can freely vary, for example, by allowing a single particle to move across a landscape with varying coupling strength, it will come to a state with the lowest vacuum energy even if the system is not coherently or thermally excited.

These theoretical values follow the experimentally obtained trend (circles), which was obtained using the measured cavity and polariton energies with only the bare plasmon frequency $\omega_{pl}$ adopted from the fitting. The theory predicts a relatively slow dependence of the normalized ground-state energy change on the detuning, whereas the experiment is more sensitive to that. This can be explained by the error in the determination of polaritons energies: in particular, the UP energy has been extracted with the

error of up to ±0.1 eV, which already constitutes a few percent of the total ground state energy. Calculating the difference of two close values $\tilde{E}_G - E_G$ makes the relative error even worse. An additional possible source of disagreement is that the true polariton energies do not exactly correspond to the extrema of a response function, such as transmission or reflection, but lie rather close to them due to the multi-mode nature of the system and the Fano resonance mechanism[6]. Despite the non-ideal agreement, however, we stress that both theoretical predictions and experimental reflectivity data signal the ground-state energy modification of the order of 10% in our plasmon–microcavity systems. Such a modification is a clear hallmark of ultrastrong coupling, since in the conventional strong coupling picture, where $g \ll \omega$, the additive coupled and uncoupled energies are exactly the same, i.e., $\omega_+ + \omega_- = \omega_{cav} + \omega_{pl}$, as can be seen from the Jaynes–Cummings model.

Lastly, we study the photonic occupancy $\tilde{n}_{\mathrm{phot}} = \langle \tilde{G} | \hat{a}^\dagger \hat{a} | \tilde{G} \rangle$ of the modified ground state $|\tilde{G}\rangle$ (the plasmon occupancy of the ground state $\langle \tilde{G} | \hat{b}^\dagger \hat{b} | \tilde{G} \rangle$ equals the photonic one[15]). In the USC regime, the ground state of the system acquires a non-zero photonic component due to the aforementioned admixing of states with different excitation numbers[15]. The photonic occupancy calculated with the use of the extracted coupling strength for the $L_{\mathrm{rod}} = 300$ nm coupled systems, shown in Fig. 5b, suggests that the ground state of the ultrastrongly coupled system may contain up to 0.06 bare cavity photons for cavities resonant with the nanorod array ($\omega_{pl} = \omega_{cav} \sim 0.5$ eV). This is smaller than 0.37 photons estimated for Landau polaritons in the THz range[21], but it is still a feasible number for converting to real photons by fast modulation of the coupling strength. The photonic occupancies calculated for other plasmonic nanorods predict almost identical values, Supplementary Fig. 19.

## Discussion

Above we have presented the ground-state modification taking into consideration only the normal incidence ($k_\parallel = 0$) mode of the cavity, whereas in reality all cavity modes having various in-plane momenta $k_\parallel$ as well as TM and TE polarizations will couple to the nanorod array. Due to the periodicity of the system, modes with different $k_\parallel$ do not interact and can be treated with independent Hamiltonians. The full vacuum energy per unit area of the cavity therefore can be calculated by integrating the vacuum energy over the entire $k$-space of the system. However, such an integration will diverge due to the asymptotic growth of the Fabry–Pérot modes energy at large $k_\parallel$. A regularization scheme will likely be needed to obtain a finite value similarly to the well-known result of Casimir[42]; these calculations will be considered elsewhere.

Our plasmon–microcavity system offers a number of interesting perspectives. First, we have studied coupled systems with only one layer of plasmonic nanoparticles that occupies only ~4% of the cavity interior. However, one can readily scale up the process and place several plasmonic layers in the center of the cavity close to the electric field anti-node. For example, placing four identical layers, assuming they all interact with the maximal electric field, will double the coupling strength and enable deep ultrastrong coupling with $g_C/\omega_{pl} > 1$. One can also note from Fig. 5b that the extracted value of $g_C/\omega_{pl}$ monotonically increases with the nanorod length in the range of studied parameters. It is therefore interesting whether the normalized coupling strength can be further boosted by increasing the nanorods length, and at which rod length the maximal $g_C/\omega_{pl}$ ratio can be expected? As we showed previously, $g_C/\omega_{pl}$ in our system scales as $\mu_{pl}\sqrt{\rho}$, which likely has an optimum. Furthermore, by precisely controlling the

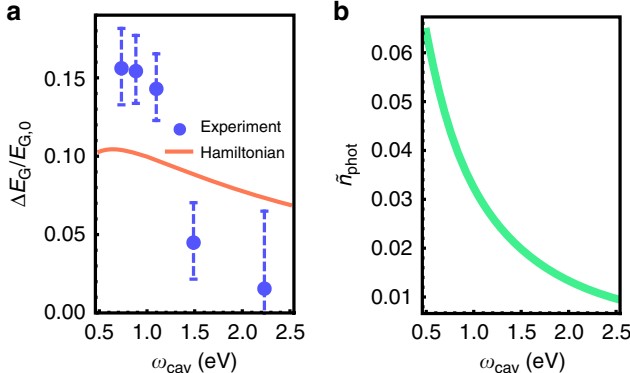

**Fig. 5 Modification of the vacuum state by the ultrastrong coupling. a** The normalized vacuum energy variation in the coupled plasmon–cavity system as a function of the bare cavity energy for normal incidence eigenmodes calculated with the coupling strength obtained from fitting of the $L_{\mathrm{rod}} = 300$ nm system, as well as the vacuum energy variation calculated directly from the measured polariton energies (error bars show 95% confidence intervals for indirect measurements). **b** The photonic occupancy of the modified ground state in the coupled system as a function of the bare cavity energy calculated with the coupling strength obtained from fitting of the $L_{\mathrm{rod}} = 300$ nm system.

nanoparticles density, our system allows creating a vacuum energy gradient in the lateral direction. Lastly, the nanoparticles can be made chiral[43], opening the opportunities to create chiral vacuum states with various vacuum energies depending on the handedness of the chiral meta-atom.

To conclude, we have demonstrated a room-temperature ultrastrong coupling between two optical harmonic oscillators: a Fabry–Pérot microcavity and an array of plasmonic nanorods. The coupling strength reaches more than half of the cavity transition energy, thus unambiguously indicating the USC regime and one of the highest values for room-temperature implementations of $g_C/\omega_{pl} > 0.55$. Importantly, this high value is achieved by filling only about 4% of the mode interior by plasmonic nanostructures, as opposed to alternative room temperature realization such as organic molecules and intersubband polaritons, which reach USC at 100% cavity filling factor. Analysis of the experimental data with the use of the Hopfield Hamiltonian reveals significant deviation of the coupled system's eigenenergies from those predicted by the naive coupled oscillators model. Remarkably, the naive models fail to describe our system despite its obvious classical nature—both system's components, plasmonic arrays, and Au mirrors contain millions of electrons and thus can be treated as classical harmonic oscillators. Furthermore, we indirectly observed a modification of the ground-state energy (up to 10%) and associated with that finite photonic occupancy induced by the ultrastrong coupling. We expect this vacuum energy change to be observable directly by the action of the cavity vacuum potential on a single nanoparticle, or via the dynamical Casimir effect. Our findings thus introduce a promising platform for studies of USC and related phenomena in the optical and infrared range at ambient conditions.

## Methods

**Samples fabrication**. All samples were prepared on thin microscope glass (170 μm) coverslips. The glass coverslips were cleaned in acetone and isopropanol at 60 °C in ultrasonicator, dried with N$_2$ blow, followed by oxygen plasma cleaning. Subsequently, 10 nm of gold (Au) mirror was prepared by e-beam evaporator with adhesion layer of chromium (2 nm) to form a bottom mirror. Then, various thicknesses of SiO$_2$ layer for half-cavities were deposited by plasma-enhanced chemical vapor deposition (PECVD at 300 °C) on top of a freshly-prepared bottom gold mirror.

To fabricate a coupled system, lattice arrays of gold nanorods with various sizes and densities were fabricated on top of the half-cavities using a standard e-beam lithography. Then, the top-half SiO$_2$ layers with the same thicknesses as the bottom SiO$_2$ half cavities were deposited using PECVD. Finally, the coupled samples were completed by a deposition of 10 nm gold film as a top mirror for Fabry–Pérot cavity. Bare nanorod samples were prepared directly on top of glass substrates as a reference sample. To perform further SEM characterization, the samples were coated by a thin layer of conductive polymer (E-spacer). Morphology of the samples was characterized using a Zeiss (Germany) scanning electron microscope (SEM ULTRA 55 FEG).

**Optical measurements**. Infrared optical measurements were performed with a Bruker Hyperion 2000 IR microscope (Schwarzschild-objective with 15× magnification, NA = 0.4) coupled to a Fourier-transform Bruker Vertex 80v spectrometer with a liquid-nitrogen-cooled mercury cadmium telluride detector. Reflection and transmission spectra were collected at normal incidence from a sample area of about 80 × 80 μm$^2$ with 2 cm$^{-1}$ resolution. All spectra were obtained with CaF$_2$ IR polarizer in two principle orientations with the electric field polarization parallel and perpendicular to the nanorods long axis. A plane gold mirror was used as a reference in the reflection configuration experiment. Broad band absorption spectra were calculated from the measured reflection and transmission spectra. Reflection spectra in visible spectrum range were collected at normal incidence using a 20× magnification objective (Nikon, NA = 0.45), directed to a fiber-coupled spectrometer and normalized with reflection from a standard dielectric-coated silver mirror.

**FDTD simulations**. FDTD simulations of the electromagnetic response of the coupled plasmon–cavity system were performed using commercial software (FDTD Solutions, Lumerical, Inc., Canada). Transmission and absorption spectra, as well as electromagnetic field distributions, were obtained with the use of a linearly polarized normally incident plane wave source and periodic boundary conditions with symmetries. The plane wave was polarized either along the nanorods or perpendicular to them. The permittivity of gold was approximated by interpolating the experimental data from Palik in the range 600–8000 nm. The simulation volume was discretized into a $\Delta r = 4$ nm mesh with further refinement of 2 nm around the metal structures (nanorod and both mirrors).

**Hopfield Hamiltonian diagonalization**. Spectrum of transition energies of Hamiltonian (1) with the interaction part (2) can be obtained as solutions of the following eigenproblem[34]:

$$[\hat{H}, \hat{P}] = \hbar\omega_\pm \hat{P}, \qquad (4)$$

where $\hat{P} = \alpha\hat{a} + \beta\hat{b} + \gamma\hat{a}^\dagger + \delta\hat{b}^\dagger$ is the polariton operator. Rewriting the eigenproblem in the basis of $\hat{a}$, $\hat{b}$, $\hat{a}^\dagger$, and $\hat{b}^\dagger$, solutions can be found as eigenvalues of the Hopfield matrix:

$$\hat{M} = \begin{pmatrix} \omega_{cav} + 2\frac{g_C^2}{\omega_{pl}} & -2\frac{g_C^2}{\omega_{pl}} & -ig_C & -ig_C \\ 2\frac{g_C^2}{\omega_{pl}} & -\omega_{cav} - 2\frac{g_C^2}{\omega_{pl}} & -ig_C & -ig_C \\ ig_C & -ig_C & \omega_{pl} & 0 \\ -ig_C & ig_C & 0 & -\omega_{pl} \end{pmatrix} \qquad (5)$$

Two eigenvalues $\omega_\pm$ of the above matrix are given by the positive solutions of the bi-quadratic equation:

$$\left(\omega_\pm^2 - \omega_{cav}^2\right)\left(\omega_\pm^2 - \omega_{pl}^2\right) - \frac{4g_C^2\omega_\pm^2\omega_{cav}}{\omega_{pl}} = 0 \qquad (6)$$

Thanks to the harmonicity of the coupled system, its entire energy ladder can be restored by collecting all possible values $\omega_{n,m} = \omega_0 + n\omega_+ + m\omega_-$ where $n$ and $m$ are non-negative integers and $\omega_0 = \frac{\omega_+ + \omega_-}{2}$.

## Data availability
The data supporting the findings of this study, including experimental reflection and absorption spectra of samples, extracted polariton energies, the Mathematica code used for the data analysis, and simulated electric field plots can be downloaded at https://doi.org/10.5281/zenodo.3727173. Additional data are provided in Supplementary Information and are available from the authors upon reasonable request.

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

## Acknowledgements

D.G.B., B.M., A.C., B.R., G.J., and T.S. acknowledge financial support from Swedish Research Council (VR Miljö grant 2016-06059 and VR project grant 2017-04545). E.Z. acknowledges financial support from RSF (Optical measurements of the structures were performed with the financial support of the Grant No. 17-79-20418). T.J.A. acknowledges support from the Polish National Science Center via the project 2017/25/B/ST3/00744. Open access funding provided by Chalmers University of Technology.

## Author contributions

D.G.B. performed Hopfield Hamiltonian analysis. B.M. prepared the samples. E.Z. and B.M. performed FTIR and optical reflectivity measurements. A.B and A.C. prepared initial samples and performed initial simulations. T.J.A. and D.G.B. performed numerical FDTD simulations. D.G.B. and B.R. analyzed the data. D.G.B., T.J.A., and T.S. wrote the manuscript with input from all co-authors. G.J. and T.S. supervised the project. All authors discussed the results.

## Competing interests

The authors declare no competing interests.
