## [Peer Review File · Nature Communications]

Reviewers' comments:

Reviewer #1 (Remarks to the Author):

In this work the authors investigate an array of plasmonic nanorods coupled to modes of a Fabry-Perot microcavity. They consider in detail the coupling between the lowest plasmonic excitation of the nanorod array and the lowest Fabry-Perot mode. They find an anti-crossing of the two excitations with a Rabi splitting that would correspond to a value of coupling strength (g) over excitation energy (ω) of the order $g/\omega \sim 0.55$ if compared to a simple model. This value is remarkable insofar that it is achieved at ambient conditions and with a minimal filling factor of the cavity (roughly 4%) and is hence one of the highest values attained at room temperature.

The presented results are very interesting and highlight the capabilities of plasmonic systems to achieve ultrastrong light-matter coupling at ambient conditions. Due to many recent applications where (ultra)strong-coupling is used to modify chemical and physical properties of material subsystems, the work is also very timely and is sure to have an impact on the highly interdisciplinary scientific community working on strong light-matter coupling.

The impact is clear in terms of establishing coupled plasmon-microcavity systems as a platform to achieve ultrastrong coupling at room temperature. But I think that the results could also have an impact on a more fundamental level and could provide further insight into the nature of (ultra)strong coupling and how it can modify material properties. To be more specific, the present results could help to clarify to what extent the collective (ultra)strong coupling observed in many different situations is a classical or a quantum effect. On the one hand the authors write about the "[...] obvious classical nature [...]" of their system and provide seemingly accurate results of a purely classical finite-difference time-domain (FDTD) simulation, on the other hand they employ a quantum model to interpret their experiment. While this allows them to show that the usual Jaynes-Cummings or Dicke models fail, the employed model of Eq.~(1) by construction provides the same (linear-response) spectrum as its classical counter part (see "Discussion" below). From this perspective I do not understand why the authors need to invoke a quantum model to interpret their results.

Therefore I would like to ask the authors to clarify to what extent they consider the observed ultrastrong coupling phenomenon a classical or quantum effect? Connected to this I would also like them to elucidate on whether they use the quantum model as only a simple tool to get a rough idea on how the coupling affects the equilibrium system or whether they consider it a genuine macroscopic quantum system at ambient conditions?

While this question is not of major relevance for the existence (and the induced effects) of (ultra)strong coupling at room temperature, which is the topic of the present manuscript, a more in depth discussion could stimulate much further experimental and theoretical work.

Discussion:

Let me briefly explain why I would like the authors to clarify their perspective and why I am not absolutely convinced about their line of arguments.

In the current manuscript the authors refer mainly to (ultra)strong-coupling situations in the quantum regime, e.g., Landau polaritons. They then use the minimal-coupling Hamiltonian (in single mode and dipole approximation) in its quantized form in Eq.~(1) to analyze their results. They argue that they do so in order to (on line 151) "[...] perform a more rigorous estimate [...]" of the coupling strength. They fit their quantum model to the measured spectra and then provide estimates on changes in the ground-state energies and the ground-state photon occupations of the nanorod-array coupled to the microcavity at ambient conditions.

However, besides the fact that they use classical FDTD simulations for fitting parameters of the quantum model (see Note S2), they show spectra obtained from classical FDTD simulations for the coupled nanorod-microcavity system (in Figs.~1.b and S1-S5) that seem to be in very good quantitative agreement without any fitting. Provided I do not overlook stark discrepancies between the FDTD simulations and the experiment, I would conclude from these results that the ultrastrong-coupling effect is mainly classical and a quantum description is unnecessary and potentially misleading.

Currently, the only explanation for why the authors nevertheless consider a quantum model I infer from the introduction where the authors state in the context of classical modeling of the system that (on line 36) "[...] the coupling is mediated by the bi-linear interaction term [...]" and (on line 51) that "[...] also classical harmonic oscillators in the regime of ultrastrong coupling require description using the full Hamiltonian." Both statements could be misinterpreted to imply that a quantum description is needed. In the context of classical physics, charged particles coupled to electromagnetic fields (in the non-relativistic limit) can be described by the Abraham model (see [1-3] or for a general overview [4]), which in its Hamiltonian form (see Eq.~(13.24) in [4]) takes the well-known minimal coupling form

$$1/2m(p-e/c A(x))^2 + v(x) + H_{\text{elmag}},$$

where H_{elmag} is the usual (classical) electromagnetic field energy. Upon quantizing the Abraham model one arrives at the standard minimal-coupling Hamiltonian of non-relativistic quantum-electrodynamics [4]. It is therefore clear that the classical model corresponds to the full Hamiltonian and contains already terms such as the A^2 term and no excursion to the quantum theory is needed.

More importantly, in the context of the current work linear-response spectra are compared with the experimentally obtained spectra. In both, quantum and classical physics, these spectra are commonly obtained by perturbing the system weakly and then calculating the dipole response. In quantum theory the resulting evolution equation for the dipole expectation value can be connected to the eigen functions of the quantized Hamiltonian through the Lehmann representation [5]. But one gets the same spectrum by just calculating the dipole expectation value in time and then Fourier transforming. Since the model the authors use is two coupled quantum harmonic oscillators (see Eq.~(1) and use $x = 1/\sqrt{2\omega}(d + d^\dagger)$ and $p = i\sqrt{\omega/2}(d^\dagger - d)$, where d and d^\dagger are any of the employed annihilation and creation operators), the equation of motion for the dipole $x(t)$ is the same as the classical one (see Eqs~(13.127) and (13.128) of Ref.~[4]). Indeed, that is even the case for infinitely many coupled harmonic oscillators and hence a classical particle in a harmonic potential coupled to the classical Maxwell field has the same response spectrum as the fully quantized field theory [4]. Thus already the classical model corresponding to Eq.~(1) can be used to provide exactly the same coupling estimates as the quantized theory. Moreover, simplifying the FDTD simulations to dipole coupling and the plasmon excitation to a harmonic excitation will lead to the same linear-response spectrum as the quantum many-mode model of Note.S1. Consequently, from only the linear spectra the classical description seems more than enough and the presented FDTD simulations seem much more elaborate than the dipole-approximated quantum models.

Indeed, the only point where a quantum description seems to be helpful is in the considerations of the equilibrium of the coupled nanorod-microcavity system. In principle, irrespective whether the coupled system is assumed quantum or classical, at room temperature I would expect a (possibly grand-)canonical-ensemble description. Such a description would be quite challenging, irrespective of the quantum or classical nature of the system, and here the simple zero-temperature quantum model can be a reasonable alternative. Since it allows to ascribe non-zero expectation values even for the zero-temperature ground state one can model to some extent the results of a proper (grand-)canonical description. Considering that a zero-temperature quantum state can be mapped

to a statistical classical ensemble (see, for instance, Ref. [6] for a discussion in the current context) fitting the results to the quantum model of Eq. (1) would indeed provide a rough idea about the equilibrium properties of the coupled system.

From the proceeding discussion I think it is therefore important to clarify the basic setting (to which extend the results are supposed to be quantum) and also to which extend a zero-temperature quantum-model is used to describe an ensemble in the ground-state. Provided the authors share my perspective, I would propose to make clear from the start that the observed effects are mainly classical, that a simplified model of the FDTD simulations (dipole coupling, one mode approximation, plasmonic excitation as harmonic oscillator) is used to model the coupling strength and that its quantized form (Eq. (1)) is used to approximate the equilibrium ensemble. If the authors think that the observed effects are mainly quantum in nature and the equilibrium state of the nanorod-array-microcavity system is considered a genuine macroscopic quantum state, I would ask the authors to provide some further details and arguments why that should be the case.

With kind regards,

Michael Ruggenthaler

[1] Abraham M., Ann. Physik 10, 105 (1903).

[2] Sommerfeld A., Proc. R. Acad. Amsterdam 7, 346 (1904).

[3] Lorentz H.A., The Theory of Electrons and its Applications to the Phenomena of Light and Radiation Heat, 2nd edition, Reprinted by New York: Dover (1952).

[4] Spohn H., Dynamics of Charged Particles and their Radiation Field, Cambridge University Press (2004).

[5] Stefanucci G. and van Leeuwen R., Nonequilibrium many-body theory of quantum systems: a modern introduction, Cambridge University Press (2013).

[6] Hoffmann N.M., et al., Phys. Rev. A 99, 063819 (2019).

Reviewer #2 (Remarks to the Author):

The authors study an hybrid system constituted by a plasmonic nanorods array positioned at the antinode of a resonant optical planar microcavity. They show that this system reaches the ultrastrong coupling (USC) regime at room temperature. In view of the great interest in the realization of the light-matter coupling in different settings, I consider these results very interesting.

The manuscript is well written and the results appear to be correct.

In order to avoid confusion and misinterpretations, the authors should clarify what is the matter component in this system. In particular, the cavity mode couples with the localized surface plasmons which actually are light-matter hybrid quasi-particles composed by surface free electrons in nanorods dressed by localized electromagnetic fields.

As the authors clearly explain, the presented results can all be described by a classical model. In the final paragraphs, the authors should briefly describe the potential of this system for observing quantum effects and/or other interesting effects and applications, beyond the physics of two coupled harmonic oscillators.

I think that, after these improvement, this manuscript can be published on Nature Communications.

Reply to Referee 1

We are very pleased to read a highly positive and constructive response of the referee.

Comment 1. Therefore I would like to ask the authors to clarify to what extent they consider the observed ultrastrong coupling phenomenon a classical or quantum effect? Connected to this I would also like them to elucidate on whether they use the quantum model as only a simple tool to get a rough idea on how the coupling affects the equilibrium system or whether they consider it a genuine macroscopic quantum system at ambient conditions?

Our reply: We thank the referee for this comment, and for a very detailed and extensive discussion following it. The answer to this question is already mostly given by the referee, with whom we are on the same page.

Indeed, our system is fully classical in the sense that any of its linear response functions, such as reflection, extinction, or absorption, can be obtained using a classical model, which is in fact demonstrated by our FDTD simulations. The Hopfield Hamiltonian of two quantum harmonic oscillators developed back in 1958 is in essence a classical model: its energy spectrum and linear response to a weak driving are identical to those obtained from a standard classical Hamiltonian of two coupled oscillators (provided that the fast rotating and the diamagnetic terms are preserved in the Hamiltonian). Thus, the quantum Hopfield Hamiltonian, which we are using, is, essentially, the classical Hamiltonian multiplied by \hbar (h-bar) – Planck's constant. The only difference between the two come at the level of fluctuations: the quantum Hamiltonian is made up of operators with bosonic commutation relations, whereas the classical one contains c-numbers.

We utilize the quantum Hamiltonian in our manuscript for two reasons. First, it provides *from start* a convenient and illustrative second quantization picture describing the *modes amplitudes* via creation operators, as opposed to the canonical coordinate and momentum in the classical Hamiltonian (which, of course, can be transformed to the modal amplitudes, but that would require an extra step).

Second, and most important, just like the referee argues in the discussion, the use of the quantum Hamiltonian allows us not only to calculate the resonant transitions, but also to obtain the characteristics of the ground state of the system in a straightforward way. These characteristics include the ground-state photonic population and the absolute ground-state energy, which follow naturally from the quantum Hamiltonian description of the system and allow us to make predictions about the zero-temperature ground state. Importantly, such characteristics are quantum and thus cannot be obtained from Maxwell's equations directly. This is one of the most important motivations for us to use the quantum model.

Thus, our system is not a genuine macroscopic quantum system as it is classical in terms of its linear response functions, it is also classical in a sense that it cannot be used to build say a Qubit and it contains no quantum superpositions (except for those of classical harmonic oscillators). But it is genuinely quantum in the sense in which every quantum harmonic oscillator is quantum, that is, it contains unavoidable irremovable quantum fluctuations associated with its zero-point energy – $\hbar\omega/2$ (also note that at room temperature the thermal energy is significantly smaller than our system's zero-point energy). In our system, this zero-point energy is affected by USC between plasmons and microcavity photons, such that:

$\frac{\hbar}{2}(\omega_+ + \omega_-) > \frac{\hbar}{2}(\omega_{cav} + \omega_{pl})$ at normal incidence. We use it as a “rough estimate” as the reviewer mentioned. As described in the main text, the full ground-state energy should include an integral over all modes of the system (all k -vectors and all polarizations), which is a problem that goes outside of the scope of the current manuscript.

An example of where this sort of quantumness could be potentially useful includes the well-known Casimir effect (which although also can be obtained using classical methods, ultimately contains the Planck's constant) as well as its dynamical counterpart.

In order to emphasize this fact, we have revised the manuscript, and added the following paragraph on p. 10:

“We want to emphasize that although our system is essentially classical, we choose to use the quantum Hamiltonian because it provides a convenient description in terms of the modes amplitudes via creation operators from start. Moreover, the use of the quantum Hamiltonian

allows us to obtain the characteristics of the ground state of the system in a straightforward way, which we will analyze in the following. However, the linear response and the energy spectrum of the system can be equally obtained from a classical description not involving any operator algebra, which is demonstrated above by the FDTD simulations.”

Reply to Referee 2

We are very pleased to read a highly positive and constructive response of the referee.

Comment 1. In order to avoid confusion and misinterpretations, the authors should clarify what is the matter component in this system. In particular, the cavity mode couples with the localized surface plasmons which actually are light-matter hybrid quasi-particles composed by surface free electrons in nanorods dressed by localized electromagnetic fields.

Our reply: We thank the referee for this comment. Indeed, we agree with the referee that the localized surface plasmon is a quasi-particle that is the result of coupling between free electrons of the metal with the vacuum electromagnetic field. However, the free electrons in this process couple to unbounded modes of the free space, i.e. plane waves, and this coupling bears mostly dissipative (radiative) character. In our case, the plasmons are positioned between the mirrors, where there are no free space modes in the first place. Therefore, the nanorod plasmons can be rightfully treated as the matter component of the coupled system. To address this comment, we added the following remark on p. 9 before Eq. 1:

“The cavity mode plays the role of the light component of the system, whereas the nanorod plasmon mode plays the role of the matter component.”

Comment 2. As the authors clearly explain, the presented results can all be described by a classical model. In the final paragraphs, the authors should briefly describe the potential of this system for observing quantum effects and/or other interesting effects and applications, beyond the physics of two coupled harmonic oscillators.

Our reply: We thank the referee for this suggestion. The discussion and conclusion paragraph currently contains the following proposals for observation of the unique quantum nature of this system:

“Furthermore, by precisely controlling the nanoparticles density, our system allows creating a vacuum energy gradient in the lateral direction. Lastly, the nanoparticles can be made chiral⁴³, opening the opportunities to create chiral vacuum states with various vacuum energies depending on the handedness of the chiral meta-atom.”

and

“We expect this vacuum energy change to be observable directly by the action of the cavity vacuum potential on a single nanoparticle, or via the dynamical Casimir effect. Our findings thus introduce a promising platform for studies of USC and related phenomena in the optical and infrared range at ambient conditions.”

We believe this short discussion indicates the potential of this system for observation of quantum effects.

REVIEWERS' COMMENTS

Reviewer #1 (Remarks to the Author):

The authors have responded to my inquiries in detail in their rebuttal, and they have correspondingly modified their manuscript. The added clarification highlights more directly the classical nature of the observed ultra-strong coupling effects. Together with providing an argument concerning the usage of the Hopfield model to estimate the ground-state modifications, the new version of the manuscript addresses these two points appropriately.

I believe that the work is of broad interest, well written and establishes plasmon-microcavity systems as a platform to achieve ultra-strong coupling at ambient conditions. I support publication of the manuscript in its current form.

With kind regards,

Michael Ruggenthaler

Reviewer #2 (Remarks to the Author):

The authors have clarified all the issues raised by the reviewers. They also improved the manuscript accordingly.

In my opinion, the manuscript can be published in Nature Communications in the present form